# The Role of Capsule Endoscopy in the Diagnosis and Management of Small Bowel Tumors: A Narrative Review

**DOI:** 10.3390/cancers16020262

**Published:** 2024-01-07

**Authors:** Stefano Fantasia, Pablo Cortegoso Valdivia, Stefano Kayali, George Koulaouzidis, Marco Pennazio, Anastasios Koulaouzidis

**Affiliations:** 1Gastroenterology and Endoscopy Unit, University Hospital of Parma, University of Parma, 43126 Parma, Italy; stefano.fantasia@unipr.it (S.F.); stefanokayali@gmail.com (S.K.); 2Department of Medicine and Surgery, University of Parma, 43125 Parma, Italy; 3Department of Biochemical Sciences, Pomeranian Medical University, 70204 Szczecin, Poland; koulaou@yahoo.co.uk; 4University Division of Gastroenterology, City of Health and Science University Hospital, University of Turin, 10126 Turin, Italy; pennazio.marco@gmail.com; 5Department of Clinical Research, University of Southern Denmark, 5230 Odense, Denmark; akoulaouzidis@hotmail.com; 6Department of Gastroenterology, OUH Svendborg Sygehus, 5700 Svendborg, Denmark; 7Surgical Research Unit, Odense University Hospital, 5000 Odense, Denmark; 8Department of Social Medicine and Public Health, Pomeranian Medical University, 70204 Szczecin, Poland

**Keywords:** small bowel tumor, capsule endoscopy, neoplasm, imaging, diagnosis

## Abstract

**Simple Summary:**

Small bowel tumors (SBTs) are rare entities with a steadily rising incidence in the last decades. The introduction of small bowel capsule endoscopy (SBCE) at the dawn of the new millennium represented a milestone in the early diagnosis and management of SBTs: this game-changing tool allowed clinicians to explore and manage pathology in previously inaccessible tracts of the digestive system. SBTs, encompassing various histological types, pose diagnostic challenges due to their rarity and non-specific clinical presentation. Early detection, crucial for prognosis, often requires a combination of diagnostic techniques, including cross-sectional imaging. With this narrative review, we aim to provide an updated insight into the significance of SBCE in addressing the complex management of SBTs.

**Abstract:**

Small bowel tumors (SBT) are relatively rare, but have had a steadily increasing incidence in the last few decades. Small bowel capsule endoscopy (SBCE) and device-assisted enteroscopy are the main endoscopic techniques for the study of the small bowel, the latter additionally providing sampling and therapeutic options, and hence acting complementary to SBCE in the diagnostic work-up. Although a single diagnostic modality is often insufficient in the setting of SBTs, SBCE is a fundamental tool to drive further management towards a definitive diagnosis. The aim of this paper is to provide a concise narrative review of the role of SBCE in the diagnosis and management of SBTs.

## 1. Background and Definitions

The advent and spread of small bowel (SB) endoscopy in the early 2000s marked the beginning of a new era, as novel devices such as small bowel capsule endoscopy (SBCE) and device-assisted enteroscopy (DAE) allowed clinicians to explore, diagnose, and treat a wide spectrum of SB pathologies traditionally out of reach of conventional endoscopy. SBCE is a non-invasive modality that allows the detection and diagnosis of SB lesions; on the other hand, DAE, which includes double-balloon enteroscopy (DBE), single-balloon enteroscopy (SBE), spiral enteroscopy, and balloon-guided enteroscopy, allows tissue sampling and therapeutic maneuvers. SBCE and DAE are complementary tools in the clinical management of SB pathologies [1].

Small bowel tumors (SBTs) are relatively rare lesions that may be found in the gastrointestinal (GI) tract, with a variable incidence in the digestive segment according to their histological type—to date, up to forty histological types have been classified, both benign and malignant [2]. The most common primary malignant lesions are neuroendocrine tumors (NETs), followed by adenocarcinomas, lymphomas, and sarcomas (among which, GI stromal tumor (GIST) is the most frequent) [3]. Although the prevalence of subtypes may vary geographically [4], the rarity and non-specific subtle clinical presentation of these entities often make the diagnostic pathway extremely challenging. The early detection of SBTs is of paramount importance to improve the patient’s prognosis, keeping in mind that in most cases a definitive diagnosis is obtained using a combination of several diagnostic techniques.

The purpose of this paper is to provide an updated overview of the role of SBCE, besides other diagnostic techniques, in the diagnosis and management of SBTs.

### 1.1. Capsule Endoscopy

SBCE was introduced in clinical practice in 2001, and rapidly became the master device for direct exploration of prior unreachable segments of the GI tract [5].

Nowadays, several types of capsules from various manufacturers are available on the market [6]; regardless of specific features, all of them share three main components: (i) a capsule endoscope, (ii) a wireless data recorder, and (iii) dedicated software for image review and interpretation [7].

Given its small size, the capsule is swallowed by the patient, and, without requiring air lumen inflation, slides through the gut propelled by physiological peristalsis. Meanwhile, a wireless device records all images captured by the capsule’s cameras during the GI transit for a time span of 8 to 12 h (according to the capsule type). Eventually, thanks to dedicated software, the video timeline is analyzed by the endoscopist for accurate reporting [7].

SBCE necessitates specific preparatory steps, including ingesting purgative and antifoaming agents, to optimize visual clarity. Real-time viewing is crucial in addressing potential gastric capsule retention and enabling timely interventions, such as administering prokinetic agents or guided capsule delivery into the duodenum. Following capsule ingestion, patients observe a fasting period before gradually introducing clear liquids after 2 h and solid food after 4 h. Automated software can assist in evaluating diffuse mucosal conditions, complementing, but not replacing, conventional reading methods. To ensure a comprehensive assessment, standardized scoring systems, and transit-time indices are recommended for reporting findings and locating lesions [8].

Technical specifications of marketed SB capsules from different manufacturers are available in Appendix A.

### 1.2. Small Bowel Tumors

SBTs are rare entities representing no more than 5% of neoplasms affecting the GI tract [6], with an increasing incidence in the last decades. One possible explanation for this phenomenon, although without clear evidence, could be the widespread implementation of SB endoscopy, together with cross-sectional imaging techniques, since the early 2000s [2,3,9,10,11,12,13]. It must be remembered that endoscopic appearance alone cannot reliably discriminate between malignant and benign SBTs.

### 1.3. Malignant Small Bowel Tumors

NETs are the most frequent primary malignant SBT of the SB (regardless of individual anatomical segments) according to the National Cancer Data Base [14,15]. When considering individual SB sections, NETs remain the most frequent lesions in the ileum [16] (Table 1), while adenocarcinoma is the most prevalent malignant lesion in the duodenum [17,18], the latter with a rising trend in incidence, possibly attributed to the refinement of diagnostic techniques [19].

Although the average age of onset for SBTs is 65 years [20], clinical suspicion of SBT must be kept in mind in young patients under the age of 50, especially in the setting of persistent SB bleeding or iron-deficiency anemia of unknown origin [21,22].

Moreover, metastases of the SB need to be suspected in cases of a known primary malignant tumor possibly involving the small intestine through several mechanisms, such as contiguous spread (e.g., in the case of peritoneal carcinomatosis) or hematological spread (as for melanoma, sarcoma, and solid tumors affecting the lung, breast, cervix, and colon) [23,24,25].

### 1.4. Benign Small Bowel Tumors

Within the category of SBTs, benign lesions account for approximately 40% of total lesions. Benign SBTs often remain asymptomatic for years, and may become clinically manifest due to iron-deficiency anemia, GI bleeding (either overt or occult, often intermittent), or abdominal pain; possible described complications are obstruction, perforation, or intussusception. The most common benign lesions are adenomas, which alone justify one-third of the number of benign SBTs, followed in order of frequency by lipomas, leiomyomas, and hemangiomas, all sharing a mesenchymal origin. The lowest in terms of frequency are inflammatory and hamartomatous polyps, the latter mostly being found in the setting of hereditary polyposis syndrome (HPS) [26,27].

Benign SBTs generally affect distal SB segments with a higher frequency (proximal-to-distal increasing trend) [28]. Conversely, adenomas are mostly found in proximal segments (i.e., the duodenum) [27].

## 2. Diagnosis

### 2.1. When to Suspect a Small Bowel Tumor

Considering the rarity of SBTs and the lack of specific symptoms (e.g., abdominal pain, anorexia, weight loss, and obstructive symptoms), their detection can be incidental; therefore, a high grade of suspicion is required to obtain an early diagnosis [3]. A common clinical scenario for SBT diagnosis is suspected SB bleeding or unexplained iron-deficiency anemia, with a diagnosis of up to 5% of all procedures performed using these indications (3% if only considering malignant SBTs) [29]. It should be emphasized that this percentage also includes individuals under the age of 50, in whom an undefined scenario of iron-deficiency anemia should prompt consideration of this diagnostic possibility by raising the index of suspicion [21,22].

The European Society of Gastrointestinal Endoscopy (ESGE) guidelines identify a category of clinical scenarios deemed to be high risks of developing SBTs and for which the use of SBCE is recommended. This includes the presence of liver metastases from a previously undiagnosed primary NET, stage III–IV malignant melanoma, and complicated/non-responsive/refractory celiac disease [3,29,30,31,32,33,34,35]. Finally, SBTs must be ruled out (when deemed clinically appropriate, e.g., in cases of suspected small bowel bleeding or unexplained iron-deficiency anemia) in the setting of HPS, such as familial adenomatous polyposis and Peutz–Jeghers syndrome, in which adenomatous and hamartomatous polyps, as well as malignant neoplasms, can involve the small intestine [29,36]. Conversely, albeit with a similar overall risk of developing a SB malignancy [37], systematic screening using SBCE in asymptomatic Lynch syndrome patients is still a matter of debate [36].

It is important to stress that whenever clinical suspicion of SBT is high, even in a case of a negative initial examination (either SBCE or cross-sectional imaging), the clinical work-up should continue, taking into consideration the intrinsic limitations of each diagnostic technique (Figure 1).

### 2.2. Diagnostic Performance of Small Bowel Capsule Endoscopy

SBCE is known to be an excellent diagnostic tool for identifying mucosal lesions [13,38,39,40,41]. The diagnostic yield (DY) of SBCE for SBTs varies depending on the indication for the examination [39,42]: in patients with suspected small bowel bleeding or iron-deficiency anemia, it ranges from 2.4% to 8.9% [43,44]. When compared to other endoscopic techniques, SBCE outperforms push enteroscopy [45], but compared to DAE (both for SBE and DBE) it seems to have a lower DY [46,47,48]. Nevertheless, it must be kept in mind that DAE is usually performed when suspicion of a lesion has already been raised using SBCE or cross-sectional imaging. Therefore, a direct comparison in terms of DY cannot be made.

When compared to cross-sectional imaging, data are conflicting: specifically, for SBTs, some authors reported a lower diagnostic sensitivity for SBCE compared to computed tomography enterography (CTE), while others found no significant differences [49,50]. Instead, considering magnetic resonance enterography (MRE), a study by Zhang et al. found a lower sensitivity and specificity for this method compared to SBCE in the setting of SBTs [51]. It is to be noted that in specific clinical conditions (such as for Peutz–Jeghers syndrome-associated hamartomas), SBCE and MRE showed comparable sensitivity, even if SBCE maintained a higher DY for small polyps (<15 mm) [52,53].

### 2.3. Small Bowel Capsule Endoscopy Limitations

Apart from technical limitations, such as the lack of therapeutic capabilities, several issues affect the diagnostic power of SBCE.

A first, purely anatomical limitation concerns the first part of the SB (the duodenum and proximal jejunum) where the visualization of lesions can be affected by the faster passage of the capsule and the presence of abundant bile secretions staining the lumen [54]. Moreover, large masses near the ligament of Treitz may be subject to traction on the mobile jejunal loop, which can be stretched, resulting in a “miss” by the forward-view capsule during the angled passage (Figure 2).

Several reports quantify the risk of SBT oversight by SBCE in these anatomical segments to be around 20%; in a retrospective study comparing the performance of third-generation capsules (PillCam^®^ SB3) versus previous versions with lower image quality, the diagnostic rate (gold standard: DAE) in duodenum/proximal jejunum for SBTs was higher (91% vs. 72%, *p* < 0.05) [55]. The use of double-headed capsules could represent a possible solution to this issue, as the second camera may provide additional significant images, especially for lesions hidden behind narrow bends [56]; nevertheless, the evidence on the matter is still too limited to allow specific recommendations [57].

Specifically, image quality and the degree of visualization are crucial parameters affecting the diagnostic performance of SBCE: among other categories of SB lesions, malignancy suffers the most negative effect using SBCE DY in the case of poor-quality images [58].

The risk of SBCE false negatives is higher for SBTs originating in subepithelial layers, as in the case of GISTs [59] and NETs [60]. Subepithelial lesions (SELs) often appear as wall bulgings that can be misread as benign findings using SBCE [6]; therefore, alert signs during SBCE reading (such as bridging folds, stretched mucosa, altered villi on the surface, central umbilication, or surface ulceration) should always increase the suspicion level for the presence of an SBT (Figure 3). To minimize this risk, dedicated scores have been devised [61]. One of the most used is “smooth, protruding lesion index on capsule endoscopy” (SPICE) based on four criteria that, with a cut-off value > 2, allows discrimination between submucosal malignant masses and innocent bulges, with a positive predictive value of 71.4% and a negative predictive value of 94.4% [62]. Nevertheless, albeit their inclusion in the report is suggested by ESGE [7], a recent study that assessed the reproducibility and reliability of these scores showed overall scarce agreement, with intra-observer agreement being superior to inter-observer agreement [63].

Despite the low overall SBCE retention rate (2.1%) [41,64], several studies suggest an increased risk (up to 10 times higher) in patients with SBTs due to their tendency to cause lumen reductions [13,39,42,65]. In the context of suspected SBTs, ESGE guidelines do not recommend any specific investigations before SBCE unless the patient presents with occlusive symptoms (e.g., abdominal pain/distension, nausea/vomiting) or when the patient presents with a history of previous SB resection, abdominal/pelvic radiation, and chronic use of non-steroidal anti-inflammatory drugs [8,29]. On the other hand, in patients with an established diagnosis of Crohn’s disease, the retention rate appears to be doubled [41,66]. Therefore, in this context ESGE guidelines suggest patency testing before SBCE [29]. In a case of capsule retention, radiological imaging could be an initial diagnostic step to identify significant risk scenarios [29]. In the unfortunate event of SBCE retention, its recovery can occur through surgery when the presence of an SBT is almost certain [7], otherwise, through DAE in equivocal cases [67,68].

Ultimately, regardless of its significant sensitivity and high specificity, SBCE does not allow histologic validation and staging of endoscopic findings. Therefore, SBCE, DAE, and cross-sectional imaging should be used as complementary methods for the diagnosis and management of SBTs [29].

### 2.4. Complementary Diagnostic Techniques

#### 2.4.1. Device-Assisted Enteroscopy

DAE is a powerful tool that enables not only complete exploration of the SB, but also active intervention for SB lesions [69], allowing therapeutic management, targeted biopsies, and pre-surgery tattoos [8], with the ability to change therapeutic plans in up to two-thirds of SBT patients [70]. In particular, the effectiveness of DAE in obtaining histological confirmation of SBTs ranges from 60% to 100% [3]. However, as the diagnostic significance of biopsy sampling may remain uncertain for SELs [68], the use of endoscopic ultrasound during DAE has been proposed, demonstrating better characterization, especially for NETs and GISTs [71]. Nevertheless, DAE is an invasive and time-consuming procedure even in expert hands; therefore, the state-of-the-art best practice may be achieved in the setting of high-volume tertiary centers with a high level of expertise [1,8,72].

#### 2.4.2. Cross-Sectional Imaging

CTE and MRE are radiological techniques that, with luminal contrast agents and specific technical settings, can accurately detect SBTs [73]. Oral contrast agents distend the lumen, allowing the identification of both intraluminal and intraparietal growth with high sensitivity [74,75,76]. In the specific setting of SBTs, MRE may perform with values of sensitivity and specificity up to 91.6% and 97.6%, respectively [74]. In a retrospective study comparing the DY of multiple diagnostic tools, regarding performance, both CTE and MRE showed a similar sensitivity (63.7% vs. 70.6%, respectively), but CTE had a higher specificity (92.2% vs. 60.0%) [51].

One of the advantages of cross-sectional imaging over SBCE is the possibility of staging the disease, thus providing the clinician with important data for further management [8]. Regarding limitations, both CTE and MRE suffer from relatively low specificity, as postsurgical adhesions, inflammation, and bowel wall movement can mimic SBTs; additionally, they may not be well-tolerated due to the large volume of contrast agent that the patient must ingest to obtain adequate lumen distension [77,78].

In addition, positron emission tomography (PET) plays a role in diagnosis, monitoring, and detecting post-therapy recurrences in the context of malignant SBTs [79]. The choice of the radionuclide depends on the clinical suspicion: for instance, well-differentiated low-grade NETs (the majority of SB NETs) are scarcely detected using ^18^fluoro-deoxyglucose, whereas somatostatin receptor imaging (^68^Gallium-PET, preferably with simultaneous contrast CT) is more sensitive and may provide management changes in up to 30% of cases [80].

## 3. Future Perspectives

Despite continuous improvements in technology and expertise in the field of SBCE, this method still has limitations regarding the miss rate and long reading time [81]. Indeed, the process of reading and interpreting a full SBCE recording can range from 30 to 120 min, with a diagnostic accuracy that decreases after the first SBCE study [82] due to the impact of reader’s fatigue [83]. Therefore, in the last 10 years, several efforts have been made regarding the application of artificial intelligence (AI) via neural network-based solutions and the application of deep-learning algorithms to endoscopy [81]. Regarding SBCE, AI has been set to select a limited number of pathological recorded frames, allowing the endoscopist to skip frames containing normal findings, thus maintaining a high sensitivity for abnormalities [84]. Although the application of AI to SBCE is still in the preliminary stage, results for protruding lesions derived from the work of Saito et al. are encouraging, as they showed a detection rate of 98.6% after training the system with over 30,000 images [85]. All this considered, continuous research is driven to make AI systems reliable soon that match or exceed human ability both in terms of detection and characterization of lesions [84,86,87].

## 4. Conclusions

SBTs are rare pathological entities that require a high index of suspicion in the diagnostic work-up of patients with unexplained abdominal symptoms, suspected small bowel bleeding, or iron-deficiency anemia.

The indolent and non-specific clinical presentation of SBTs often leads to a diagnostic delay. Depending on their nature, SBCE has a variable DY, which is optimal for mucosal growth neoplasms and reliable for SELs. Given the limitations of SBCE, complementary diagnostic methods, such as DAE and cross-sectional imaging techniques, are crucial, not only for diagnosis when SBCE is inconclusive, but also for staging, monitoring, and addressing proper treatment. Furthermore, continuous technological progress is leading to the application of AI to SBCE, which, although in its early stages, is rapidly developing towards higher levels of efficiency and refinement.

## Figures and Tables

**Figure 1 cancers-16-00262-f001:**
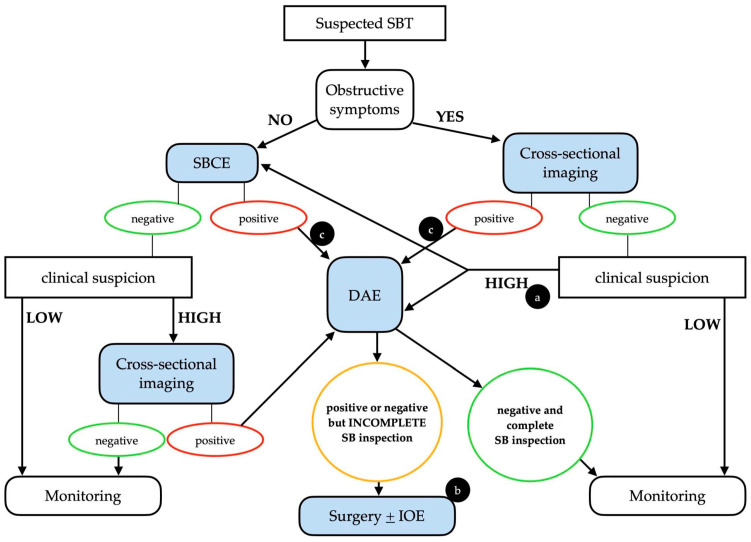
Possible clinical algorithm for the suspicion of a small bowel tumor. Legend: (a) The choice between DAE and SBCE should be made on a case-by-case basis. (b) In patients with positive SBCE and negative (but incomplete) DAE, cross-sectional imaging should be considered before surgery. (c) An upfront surgical approach may be taken into consideration in cases with clear malignant findings. DAE, device-assisted enteroscopy; IOE, intra-operative enteroscopy; SBCE, small bowel capsule endoscopy; SBT, small bowel tumor. Adapted from Rondonotti, E. et al. [3].

**Figure 2 cancers-16-00262-f002:**
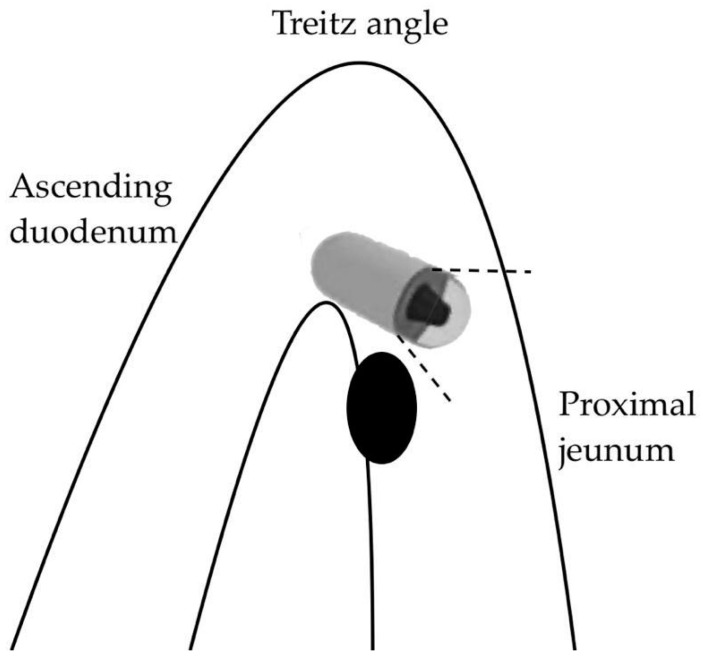
Capsule oversight of a lesion located next to the Treitz angle.

**Figure 3 cancers-16-00262-f003:**
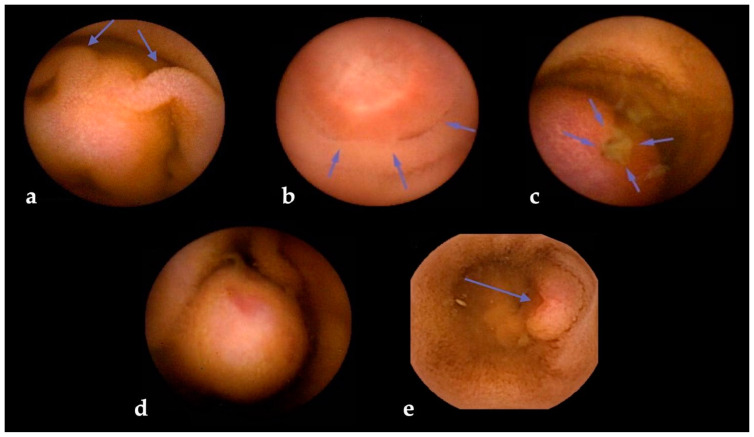
Visual clues for submucosal tumors in small bowel capsule endoscopy. Legend: (**a**) bridging mucosal folds; (**b**) mucosal stretching; (**c**) surface ulceration; (**d**) altered villi on surface; and (**e**) central umbilication.

**Table 1 cancers-16-00262-t001:** Epidemiology of primary malignant tumors of the small bowel.

Primary SB Malignant Tumors	Incidence Rate (/100,000 Person-Years) *	Mean Age at Diagnosis (Years)	Five-Year Relative Survival Rate (%)	Most Probable Location
NETs	0.83	67–68	80	Ileum
Carcinomas	0.66	67–68	28	Duodenum
Stromal tumors	0.20	60–62	58	All SB tracts
Lymphomas	0.38	60–62	64	All SB tracts

* Incidence rate is adjusted to the 2000 USA standard population, adapted from Qubaiah, O. et al. [12]. Legend: NET, neuroendocrine tumor; SB, small bowel.

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
