# Peer review of "The Role of Capsule Endoscopy in the Diagnosis and Management of Small Bowel Tumors: A Narrative Review"

_cancers, 2024, doi:10.3390/cancers16020262_

Round 1
Reviewer 1 Report
Comments and Suggestions for Authors
Comment:
This review assess the role of capsule endoscopy in the diagnosis of small bowel tumors. This review is well documented but there is some concern for several statement. There is too much abbreviation in the text that give the reading uncomfortable.
Abstract:
1) It is a non-sense to state that capsule endoscopy is the explanation of the increasing of incidence of small bowel tumor. It eventually allow an earlier diagnosis but this is not clearly supported by evidence.
Background:
1) Capsule endoscopy is not the cause of increasing incidence of small bowel tumor!
2) It is not true that metastasis is the most frequent malignant lesion of the small bowel. The reference N°16 published in 1964 is not accurate.
3) Some references are old, more recent publication may be cited (i.e. Sakae H et al, Br J Cancer 2017:1607 ; Aparicio T et al, Int J Cancer 2020:967). It must be pointed out that in Japan most of the small bowel adenocarcinoma are diagnosis with upper gastroscopy at an early stage maybe due to the screening program of gastric cancer (Sakae 2017). Capsule endoscopy is particularly useful for diagnosis of jejunum tumor (Aparicio 2020).
4) Nevertheless, despite the availability of capsule endoscopy the diagnosis of small bowel tumor did not turn to earlier stage according to Dutch registry (Legué LM et al, Acta Oncol 2016:1183)
Diagnosis:
1) It could be remembered that in young individual small bowel ulceration is more frequent than tumor.
2) According to several publication Lynch syndrome is a more frequent cause of small bowel adenocarcinoma than familial adenomatosis polyposis (Halfdanarson T et al, Am J Surg 2010:797 ; Aparicio T et al, Int J Cancer 2020:967 ; Latham A et al, Clin Cancer res 2021 1429)
3) Figure 1: surgery of a small bowel tumor without previous endoscopic diagnosis is frequent but is not propose in the algorithm
4) Capsule retention remains a challenge in case of suspicion of small bowel stenosis especially in case of inflammatory bowel disease.
Reviewer 2 Report
Comments and Suggestions for Authors
The topic of this review is interesting, but this article should be improved. Some issues should be adressed:
1. Search strategy should be added in the part after introduction.
2. Technique of SBCE should be described.
3. Complications of SBCE should be described.
4. Studies analyzing a role of SBCE in SBT diagnostics should be cited and shortly described in the text. In addition, a summary table including all studies regarding the role of SBCE in SBT diagnostics should be added. This table should contain fisrt author's name, publication year, a type of study (prospective, retrospective), number of patients, and results (including detection rate, completion rate, and retention rate).
Minor editing of English language required.
Reviewer 3 Report
Comments and Suggestions for Authors
The authors provided a clinical overview of the role of SBCE in the diagnosis and management of SBTs. The manuscript is well organized and introduces most aspects of the field of focus. However, I find the information about SBCE quite limited. In specifics:
1. From the total 8 pages only 2 pages of the manuscript a really focused to SBCE.
2. Since this is about a clinical overview, I would expect some tables summarizing main outcomes of SBCE, capsule specifications, a comparison to different capsules.
3. I would also like to see a quite larger section about the future perspectives of SBCE, also in relation to the current advances in capsule endoscopy. Again with a (some) table(s) summarizing the main studies. Is it AI only, another imaging approach besides color imaging, multimodality?
4. Finally it would be very interesting for the readers (and the technology) to include additional figures showcasing the performance of the capsule (in relation to my 2nd comment) and the future directions (in relation to my 3rd comment)
Round 2
Reviewer 2 Report
Comments and Suggestions for Authors
Current authors' responses are not sufficient. In their responses, the authors did not cited the corrected fragments of the text exactly, as well as they did not indicate in which part of the manuscript (page, paragraph, line) the corrected fragments were located. The current responses are short and perfunctory.Moreover, the authors did not mark the corrected fragments in the manuscript. Please provide a solid responses and precisely refer to which fragment was changed inthe authors' responses.
Comments on the Quality of English LanguageMinor editing of English language required.
Reviewer 3 Report
Comments and Suggestions for Authors
The authors successfully responded to the comments raised by the reviewers.
Author Response
Dear Reviewer,
thank you very much for your insight. We are grateful for your positive response.
Best regards
The authors